# Peer review of "Systematic Review and Meta-Analysis of microRNA-7-5p Expression and Biological Significance in Head and Neck Squamous Cell Carcinoma"

_cancers, 2025, doi:10.3390/cancers17193232_

Round 1

Reviewer 1 Report

Comments and Suggestions for Authors

This study conducts a systematic review and meta-analysis on the expression and biological significance of miR-7-5p in head and neck squamous cell carcinoma (HNSCC). By integrating multi-source data, it reveals the expression patterns and potential value of miR-7-5p in HNSCC. However, there are deficiencies in the design of validation experiments, mechanistic explanations, and demonstration of clinical significance.

  1. Limitations of validation experiments
    The authors validated the upregulation trend of miR-7-5p in tumors through their own clinical cohort (n=3), but the extremely small sample size and the lack of statistical significance in the correlation of target gene expression result in weak persuasiveness of the validation findings, which fail to fully support the conclusions of the meta-analysis.
  2. Indirect and insufficient evidence for mechanistic research
    Regarding the paradox that "miR-7-5p is upregulated yet exhibits tumor-suppressive functions," the authors speculate it as a "compensatory mechanism or cellular stress response" but lack direct experimental evidence:
  • In the 4NQO treatment model, acute treatment upregulates miR-7-5p while long-term treatment shows no significant difference, which only indirectly suggests its association with stress but cannot directly prove the compensatory mechanism.
  • Functional experiments (such as detecting tumor phenotypic changes after knocking out/overexpressing miR-7-5p) were not performed to verify the causal relationship between its tumor-suppressive function and upregulated expression in HNSCC.

It is suggested that this study should further improve the understanding of the biological functions and clinical application potential of miR-7-5p by expanding the sample size and conducting mechanistic validation experiments.

Comments on the Quality of English Language

This article uses words accurately, is relatively fluent, and has correct spelling. However, some sentence structures are not concise enough, and it is suggested to further improve the word order and grammar.

Reviewer 2 Report

Comments and Suggestions for Authors

This study conducted by Brown et al., underwent a systematic evaluation of expression and biological relevance of microRNA-7-5p (miR-7-5p) - especially as a risk factor for head and neck squamous cell carcinoma (HNSCC) - and using multiple public databases at the time of writing, miR-7-5p was over-expressed (or upregulated) in HNSCC tumour tissues, above all oral cancer, while likely conferring tumour suppressive activity. This detailed evaluation aimed to provide clarity about the contradictory roles of miR-7-5p, to help determine its potential context-dependent ability as diagnostic biomarker or drug target for HNSCC, and illustrated the importance of understanding context-dependent effects microbiota - especially small RNAs like miR-7-5p - for RNA-targeted therapies.

To the best of my knowledge, the innovations of this study mainly include the following points:

  1. This is the first meta-analysis of miR-7-5p expression in cancer, since no studies included in the meta-analysis solely evaluated miR-7-5p, which is remarkable that the conclusions are more definitive than any single study.

  1. The study utilized a comprehensive integration or patient-based approach from several resources to explain the clinical significance and biological function of miR-7-5p in head and neck squamous cell carcinoma (HNSCC). The integrated patient data, bioinformatics predictions, and laboratory experiments improved our understanding of the role of miR-7-5p in cancer biology and provided a rationale for a new type of RNA-based therapy in head and neck cancer.

  1. This study provides insights into the paradoxical role of miR-7-5p in HNSCC: despite its frequent upregulation in tumors and association with poor clinical outcomes, synthetic miR-7-5p mimics exhibit anticancer functions. This study provides new insights into this complex function, which may be related to cellular context and regulatory feedback mechanisms. Besides, the study highlights site-specific trends, particularly in oral cancer, suggesting the need to study oral cancer as a separate subtype.

  1. This study provides evidence that miR-7-5p could be established as a potential biomarker and therapeutic target for HNSCC. The high expression of miR-7-5p in OSCC adds to its ability to serve as a tumor-associated biomarker and as an aid to facilitate early diagnosis and tumor classification.

Some issues are raised regarding this report:

  1. Dual Roles and Paradoxes of miR-7-5p: Although synthetic miR-7-5p mimics have had anti-cancer effects, the vast literature overwhelmingly shows that miR-7-5p is often upregulated in tumors in vivo and that upregulation is associated with poor clinical outcomes. This paradox is significant as miR-7-5p has at times been described as a tumor suppressor or an oncogenic miRNA and its transciptomic gene pollution context in the HNSCC modality remains largely insufficiently addressed. The most reasonable explanation in the literature for upregulation is that it is either compensatory, in other words a cellular stress response to a redesign cellular growth limitation in response to tunmogenesis or cellular ageing or simply limited cause of oncogenic activity. In response to this contradictory phenomenon, the author proposed a hypothesis to explain it, but the experimental verification could not prove the hypothesis.

  1. Because the selection criteria excluded datasets from liquid biopsies, the results of the meta-analysis cannot be generalized to non-solid tissue samples (e.g., saliva, serum), although other studies have shown that miR-7-5p has diagnostic potential in these samples. Besides, when validating the meta-analysis results, the research team had a small number of patient samples in their own clinical cohort (n=3), which resulted in the RT-qPCR analysis results of hub genes failing to reach statistical significance.

  1. The TCGA data analysis indicated no significant association with overall survival or disease-free interval, although it was significant for disease-specific survival and progression-free interval. Although some studies have suggested that miR-7-5p may be associated with p53 mutations, the TCGA data analysis also does not show any clear relationship between miR-7-5p expression and p53 status. The authors could consider discussing the possible basis for these observations of lacking clear correlation with clinical prognosis or inconsistency with other studies, in the Discussion section.

  1. The first meta-analysis did show a substantial degree of heterogeneity regarding miR-7-5p expression (I²=76.0%). It is important to note that the study comparatively reduced heterogeneity in expression by excluding the "quantile-normalized" studies (I²=56.2%), but it was not completely eliminated. The authors may consider additional possible explanations next, regarding the relatively large heterogeneity in direction of expression, in the Discussion section.

The following are some suggestions to further enhance the depth and impact of this research:

  1. The exploration into the paradoxical mechanisms of action of miR-7-5p: Future studies may address the mechanisms of molecular underpinnings of increased expression, and whether it is due to activity of transcription factors, any epigenetic modification of the target, or the other surrounding cell types in the tumor microenvironment (fibroblasts and immune cells).

  1. Further studies regarding early diagnosis and potential precancerous lesions: While liquid biopsy samples were excluded in this meta-analysis, in the discussion, miR-7-5p was significantly increased in the saliva samples of individuals with oral squamous cell carcinoma (OSCC) and could be considered for early diagnostic biomarker use for potential oral malignancies. The field of HNSCC has no early diagnostic markers, so the researchers should focus on systematically assessing miR-7-5p expression in precancerous lesions or high-risk cohort studies, as well as its stability and predictive ability in liquid biopsies, which may encourage early screening and intervention.

  1. Improving heterogeneity in meta-analysis: While heterogeneity was significantly reduced (I² value reduced from 76.0% to 56.2%) by using subgroup and meta-regression analyses and excluding quantile normalization studies, moderate heterogeneity remained. It is recommended that whenever designing or selecting included studies, that testing methods, standardization strategies, and tumor subtype classifications are standardized, or utilize advanced statistical models to explain or mitigate the impact of residual heterogeneity making the results more generalizable

  1. Further focus on the relationship between miR-7-5p and TP53 mutation status: The article notes that the correlation between miR-7-5p levels and TP53 mutation status remains unclear and contradicts other studies. Given the important role of TP53 in HNSCC, it is recommended to investigate or discuss the expression and function of miR-7-5p and its precursor miRNAs (miR-7 pre-miRs) in tumors with different TP53 statuses, and to clarify whether p53 directly or indirectly regulates miR-7-5p, in order to clarify this complex regulatory relationship.

Here are some examples where grammar or writing can be improved:

  1. line 107, "exogenous delivery of miR-7-5p has been shown to suppresses tumor growth" should be changed to "exogenous delivery of miR-7-5p has been shown to suppress tumor growth".

  1. line 154, "miR-7-5p not detected in >60% of samples" should be changed to "miR-7-5p was not detected in >60% of samples"

  1. line 154, The sentence "data was uninterpretable it was not included" is a bit loose in structure. To make it clearer, it could change to "or if data was uninterpretable, it was not included."

  1. line 205, "real-time quantitative PCR (RT-qPCR) were conducted" should be changed to "RT-qPCR was conducted."

  1. line 215, "circos plot generated with Synteny portal" should be changed to "circos plot was generated with Synteny portal".

  1. line 245, for a period of 90 minutes" should be changed to "for a period of 90 minutes" or "for 90 minutes".

  1. line 248, "RNA harvested when cells reached" should be changed to "RNA was harvested when cells reached".

There are still additional grammar and writing errors in this report. Thorough editing is appreciated.
